# Plant Growth Inhibitory Activity and the Response of Different Rootstocks to Soil Sickness Syndrome in Japanese Pear Tree

Tomoaki Toya [1,*], Masayoshi Oshida [2], Kwame Sarpong Appiah [3,4] , Jun Takita [5] and Yoshiharu Fujii [3,*]

1 Chiba Agricultural Office, Chiba Prefectural Office, 473-2 Ookanazawa-Cho, Midori-Ku,
Chiba-City 266-0014, Japan
2 Chiba Prefectural Agriculture and Forestry Research Center, Chiba Prefectural Office, 180-1 Ookanazawa-Cho,
Midori-Ku, Chiba-City 266-0014, Japan
3 Department of International and Innovative Agriculture Science, Tokyo University of Agriculture
and Technology, 3-5-8 Saiwai-Cho, Fuchu 183-8538, Tokyo, Japan
4 Department of Crop Science, College of Basic and Applied Science, University of Ghana, Legon,
Accra P.O. Box LG 44, Ghana
5 Central Research Institute, NittoBest Corp., 4-27 Saiwai-Cho, Yamagata 991-8610, Japan
* Correspondence: t.ty5@pref.chiba.lg.jp (T.T.); yfujii@cc.tuat.ac.jp (Y.F.); Tel.: +81-080-43-3000950 (T.T.)

**Abstract:** Soil sickness syndrome in Japanese pear (*Pyrus pyrifolia* (Burm.f.) Nakai) affects the growth of the tree and decreases fruit yield. This study investigated the growth-inhibitory activity in Japanese pear (*Pyrus pyrifolia* (Burm.f.) Nakai) using the rhizosphere soil assay method to elucidate the characteristics of growth-inhibitory substances in Japanese pears. As a result, the root bark had the highest growth inhibitory activity during the growing season of the Japanese pear. For comparative analysis, the growth-inhibitory activities of Japanese apricots (*Prunus mume* Sieb. Et Zucc.) and figs (*Ficus carica* L.) were also investigated. Similar to the Japanese pear, the root bark of Japanese apricots and figs had a higher inhibition rate than the root pith. Like Japanese apricots and figs, it was inferred that the growth inhibitory substances accumulate in the bark of the Japanese pear. Furthermore, soil sickness syndrome in Japanese pear saplings did not occur when a fragment of Japanese pear shoots or thick roots was mixed with non-pear soil (soil with no history of Japanese pear cultivation). Based on these findings, it is considered that the condition of soil sickness in Japanese pear is caused by the accumulation of phenolic compounds such as arbutin, which is accumulated in the bark of the tree, secreted from the roots, and subsequently builds up in the soil. Additionally, the degree of occurrence of soil sickness syndrome depending on the rootstock was clarified. It was observed that the rate of growth inhibition was significantly higher in *Pyrus betulifolia* (Birchleaf pear) than in *Pyrus pyrifolia* (Japanese pear). Even when Japanese pear trees were planted in soils with no history of Japanese pear cultivation, the initial growth of *P. betulifolia* was 1.4 times that of *P. pyrifolia*. It is suggested that *P. betulifolia* is weak against soil sickness, but is excellent at initial growth itself. Our findings are important because *P. pyrifolia* is used for cultivation, in combination with other mitigation measures, such as soil dressing in replanted fields.

**Keywords:** growth inhibitory substances; inhibitory activity in the tree; inhibition rate of soil; rootstock





## 1. Introduction

The cultivation of the Japanese pear (*Pyrus pyrifolia* (Burm.f.) Nakai), which is a major fruit in Japan, has been expanding in East Asia. However, after 30 years of continuous cropping, the yield of Japanese pear decreases and the fruits become smaller. Therefore, many pear-producing fields are being replanted with young trees. However, the replanted saplings of Japanese pear tend to grow poorly due to soil sickness syndrome [1]. Soil sickness syndrome is caused by growth inhibitors accumulated in the soil through root leaching and root exudation by previous crops, resulting in reduced growth of subsequent crops in addition to soil disease [2]. Soil sickness syndrome is observed in many fruit trees,

such as Japanese apricot [3], fig [4], apple [5], and peach trees [6], as well as in vegetables, such as asparagus [7]. A previous study clarified that soil sickness syndrome occurs in Japanese pears and soil hardness, chemical properties, and soil diseases and pests are less involved in the causes of the occurrence [1]. Additionally, the rhizosphere soil assay method [7,8] was used to evaluate the risk of soil sickness in the continuous cropping soil of the Japanese pear [9] and also to clarify the accumulation of the growth inhibitory substances in the soil that occurred during the growth of Japanese pear trees [10]. However, it is unknown at what stage during the tree growth of Japanese and which part of the pear tree possesses high growth inhibitory effect. Therefore, in Experiment 1 of this study, we planted pear saplings in pots, took samples of each part of the tree every 2 months, and analysed them with the rhizosphere soil assay method to clarify the parts and timings where the growth inhibition reaction was strong. Additionally, in Experiment 2, to compare the results with Japanese pears, we collected and analysed samples of figs (*Ficus carica* L.) and Japanese apricot (*Prunus mume* Sieb. Et Zucc) trees, which are problematic in terms of soil sickness syndrome.

In addition, the shoot and thick root fragments of Japanese pears are left on the field during replanting. The shoot and thick roots of Japanese pear trees are generally difficult to decompose and there is a possibility that the risk of soil sickness syndrome will persist for a long time in replanted fields if growth inhibitory effects are confirmed. However, the growth inhibitory effects of the shoot and thick root of Japanese pear on saplings are still unknown. Therefore, in Experiment 3, we set up a treatment area mixed with the crushed shoots or thick roots of Japanese pear trees, planted Japanese pear saplings, and investigated the effects on tree growth.

The selection of cultivars with elevated resistance or tolerance to soil sickness syndrome is a conventional technique to minimize or reduce replanting failures. The differences in the occurrence of soil sickness among Japanese pear varieties have been reported, and this is especially noticeable in the cultivar 'Akizuki' [11]. In Japanese pears, cultivars are grafted on a strong rootstock and used for cultivation to promote growth. The most commonly used rootstocks are *P. pyrifolia* (Japanese pear), native to Japan and *P. betulifolia* (Birchleaf pear) from north-eastern China. Although drought and water resistances in *P. betulifolia* are superior to *P. pyrifolia* [12], there is no available information on tolerance to soil sickness syndrome. Therefore, in Experiment 4, we conducted a cultivation test using Japanese pear saplings grafted on *P. betulifolia* or *P. pyrifolia* using 'Akizuki' to evaluate tolerance to soil sickness syndrome.

From the above results, we clarified some of the causes of soil sickness syndrome in Japanese pears, such as the growth-inhibitory activities in the tree and the differences in the occurrence of soil sickness among the rootstocks.

## 2. Materials and Methods

### 2.1. Measurement of Growth Inhibitory Activity via the Tissue of Japanese Pear Trees (Experiment 1)

The Japanese pear trees were cultivated at the Chiba Prefectural Agriculture and Forestry Research Centre. Fifteen pots (volume 22.5 L) were filled with soil (volcanic ash soil with a high phosphate absorption coefficient and a small volumetric weight) collected from a vegetable field with no history of Japanese pear cultivation. Subsequently, Japanese pear saplings of the cultivar 'Akizuki' (*Pyrus betulifolia* Bunge) were planted on 18 March 2020. Chemical fertilizer (nitrogen: phosphorous: potassium (NPK) 15:15:15) was applied at the rate of 100 g/tree from April to July. Watering was done daily, with a nozzle installed to prevent the soil from drying out. Three trees were uprooted on 29 March 2020 (before germination), 29 May, 31 July, 30 September, and 30 November (after leaf fall). The roots were removed from the pot and washed with water.

The leaves, bark, and pith of shoots bark and pith of the main trunk, and the bark and pith of thick roots (a diameter of about 2 cm) and fine roots (2 mm or less) were divided and cut into 1 mm squares using scissors. The Japanese pear tree samples were dried at 60 °C for 4 h with a drying machine (MOV-112F, Sanyo Electric Biomedica Co. Ltd., Tokyo,



Japan) on the same day. After the trees were uprooted, the remaining soil was used as the test material (the pear soil). The soil samples were dried at 60 °C for 24 h with a drying machine, crushed, and passed through a 2 mm sieve. Sample processing was performed on the day of collection.

The rhizosphere soil assay method examines the inhibitory effects of substances released from the roots into the soil and can be used to evaluate allelopathic activity even with a small amount of soil [7,8]. Toya et al. [9] adopted the rhizosphere soil assay method to evaluate the risk of soil sickness syndrome in Japanese pears using lettuce as a test plant. In the cultivation test, when the soil inhibition rate at the time of planting was 30%, the dry matter weight was 89% of the non-pear soil plot. At a soil inhibition rate of 40%, the dry matter was 66% of the non-pear soil plot and when the inhibition rate was 60%, the dry weight was 55% of the non-pear soil plot [9].

The analysis of the tree samples was performed by adopting the rhizosphere soil assay method with some modifications, according to Motoki et al. [13]. Motoki et al. [13] used plant samples instead of soil samples to determine the inhibition rate. In this test, 100 mg of oven-dried tree samples were placed in each well of the culture multi-dish (six holes, NUNC). Five mL of 0.75% low-temperature gelled agar (Nacalai Tesque Inc., Kyoto, Japan) autoclaved at 115 °C for 15 min) was added, mixed, and hardened, and then 5 mL of agar was layered. *Lactuca sativa* seeds ('Legacy', Takii Seed Co., Ltd., Kyoto, Japan) were sown onto the agar and kept at 25 °C for 3 days under dark conditions. Lettuce was used as an indicator plant because it is susceptible to allelopathy. After this, the root length of the lettuce was measured. The measurement was performed five times in each section. A well filled with only agar was prepared and used as a blank control. The growth inhibitory activity of the tree was calculated as a percentage of the blank control, as shown in the formula below.

$$z = (x - y)/x \times 100 \tag{1}$$

where z is growth inhibitory activity (%), x is the average value of blank root length, and y is the average value of root length of the tree sample.

At the same time, the inhibitory effects of the soil samples were also evaluated using the rhizosphere soil assay [14]. Three grams of soil were placed in each well of the culture multi-dish. Other procedures were the same as for tree samples. The analysis of the tree sample and soil was performed within 3 days after sampling.

*2.2. Evaluation of Growth Inhibitory Activity of Japanese Apricot and Fig Trees (Experiment 2)*

Fig (*Ficus carica* L.) 'Masui Dauphin' (4-year-old) and Japanese apricot (*Prunus mume* Sieb. Et Zucc.) 'Shirakaga' (21-year-old) were used as the test trees. Roots and soil samples were collected from a position of about 50 cm in the horizontal direction and a depth of 10 to 40 cm, and leaves and shoots were collected from the aboveground part on 29 August 2019. A hundred grams of leaves and roots were collected from each of the three trees. The roots were partly dug up and washed with water. Shoots and thick roots were divided into the bark and pith, and the roots were divided into thick roots, with a diameter of about 2 cm, and fine roots, with a diameter of 2 mm or less. Tree samples were prepared in the same manner as in Experiment 1. The analysis was performed using the rhizosphere soil assay as in Experiment 1.

*2.3. Evaluation of the Effect of Tree Fragments Mixed with Soil on the Occurrence of Soil Sickness Syndrome in the Japanese Pear (Experiment 3)*

The cultivation test was conducted at Chiba Prefectural Agriculture and Forestry Research Center. The tree samples were collected by uprooting 'Housui' (35-year-old, *P. pyrifolia*) on 24 August 2008. Shoots with a diameter of about 3 cm, thick roots with a diameter of about 2 cm, and fine roots with a diameter of 2 mm or less were collected separately and crushed with a chipper. 800 g of fragments of the shoot, thick root, and fine root was mixed with each planting hole (length: width: depth = 40:40:40 cm) at 2 m intervals on the field (volcanic ash soil) with no history of Japanese pear cultivation, on

26 October 2008 (shoot or thick root, fine root treatment). In addition, an immiscible control plot was set up. Each of the treatments had 10 replications. Japanese pear saplings 'Akizuki' were planted on December 21. Fertilization was the same as described in Experiment 1. When the volume of the planting hole and the specific gravity of the soil are calculated from 0.7 [15], the mixing amount is 1.8% by weight concerning the soil, which is a large condition as a residue of the previous tree in the ruins of rooting. Compared to other Japanese pear cultivars, 'Akizuki' has a remarkable growth inhibition due to soil sickness syndrome [11]; hence it was used as a material for verification in this study.

Cultivation after planting was customary and in the first year of pruning, the shoot was cut to half the length. The tree shape was made into a standing tree, and management practices such as pinching and attraction were not performed. The tree growth and dismantling were investigated 2 years after planting. The shoot was surveyed for lengths of 10 cm or more. The length, the diameter at 5 cm above the base, and the number of occurrences were measured. The total elongation was the sum of the lengths of all new shoots. The main trunk diameter of the tree was measured at 10 cm above the grafted part. The raw weight of each organ of the tree was evaluated separately for a new shoot, stem, leaf, and root (below the graft).

### 2.4. Evaluation of the Effect of the Difference in Rootstock on the Degree of Occurrence of Soil Sickness Syndrome in Japanese Pear (Experiment 4)

The Japanese pear was cultivated at the Chiba Prefectural Agriculture and Forestry Research Centre. Pots (volume 22.5 L) were filled with soil (volcanic ash soil) collected from the vegetable field and Japanese pear saplings of the cultivar 'Akizuki' were planted for five months. A chemical fertilizer (nitrogen: phosphorous: potassium (NPK) 15:15:15) was applied at the rate of 100 g/tree from April to July 2019. Watering was conducted daily with a nozzle installed to avoid soil dry-out. After the trees were uprooted, the remaining soil was used as the test material (the pear soil). As the non-pear soil, the soil collected from the vegetable field without the cultivation of pear sapling was used. The saplings with 'Akizuki' as the scion and *P. betulifolia* or *P. pyrifolia* as rootstock were planted in pots on 18 March 2020. Each of the treatments had 5 replications. At the time of planting, 50 g of soil was collected from the pots of each plot and the soil inhibition rate was evaluated as described in Experiment 1. The main trunk diameter of the tree at planting was measured 10 cm above the grafted part. The tree growth survey and dismantling survey were conducted 5 months after planting. The shoot and the main trunk diameter of the tree were measured the same as in Experiment 3 on 17 August 2020. The number of leaves and leaf colour was measured. For leaf colour, 20 leaves of each tree were randomly selected and measured with a chlorophyll meter (SPAD-502, Konica Minolta Co., Ltd., Tokyo, Japan). The dry weight of each organ of the tree was evaluated the same as in Experiment 3. All samples were measured after drying at 90 °C for 1 week.

### 2.5. Statistical Analysis

The significance of the inhibitory activity and soil inhibition rate for each tissue type of the tree was tested by the Tukey–Kramer method after the angle conversion. The growth of Japanese pear trees and the weight of trees were analysed by the Tukey–Kramer method in JMP (version 5.0.1J, SAS Japan Institute Inc., Tokyo, Japan.).

## 3. Results

### 3.1. Measurement of Growth Inhibitory Activity via Tissue Type of Japanese Pear Trees (Experiment 1)

The growth inhibitory activity of thick root bark was 70.8% in July during the growing season, which was higher than the other tissue types (Table 1). In September, it was 69.6%, which was higher than 25.8% of thick root pith. There was no difference in growth-inhibitory activity depending on the tissue type of the tree in March before germination, in May during the early stage of growth, and in November after leaf fall.

**Table 1.** Changes in the growth inhibitory effects of the tissue type of Japanese pear tree.

| | Growth Inhibitory Activity of Japanese Pear Tree (%) | | | | |
|---|---|---|---|---|---|
| | **March** | **May** | **July** | **September** | **November** |
| Leaf | - | 46.2 | 38.8 a | 49.2 ab | - |
| Shoot bark | - | 38.4 | 27.8 a | 48.4 ab | 42.6 |
| Shoot pith | - | 43.8 | 22.2 a | 51.6 ab | 55.7 |
| Main trunk bark | 19.0 | 46.5 | 42.4 a | 44.2 ab | 61.4 |
| Main trunk pith | 41.0 | 22.3 | 31.4 a | 41.2 ab | 49.7 |
| Thick root bark | 37.5 | 69.3 | 70.8 b | 69.6 b | 62.0 |
| Thick root pith | 35.0 | 28.8 | 27.5 a | 25.8 a | 68.2 |
| Fine root | 42.3 | 46.2 | 42.7 a | 51.9 ab | 54.3 |
| *p*-value | 0.12 | 0.30 | <0.01 | 0.05 | 0.33 |

Data were arcsine transformed and analysed via the Tukey–Kramer method. Lowercase letters indicate a 5% level of significance between different letters.

The inhibition rate of soil in which pear trees were cultivated (pear soil) was 18.3% in March before planting, which was the same as in the non-pear soil (Figure 1). It increased to 37.9% in May and 54.3% in July as the tree grew. The highest inhibitory effect was observed in September (57.0%), followed by a decline in November (46.3%). The inhibition rates of the soil in plots without pear trees were 19.7 to 22.7%, and there was no change over time.

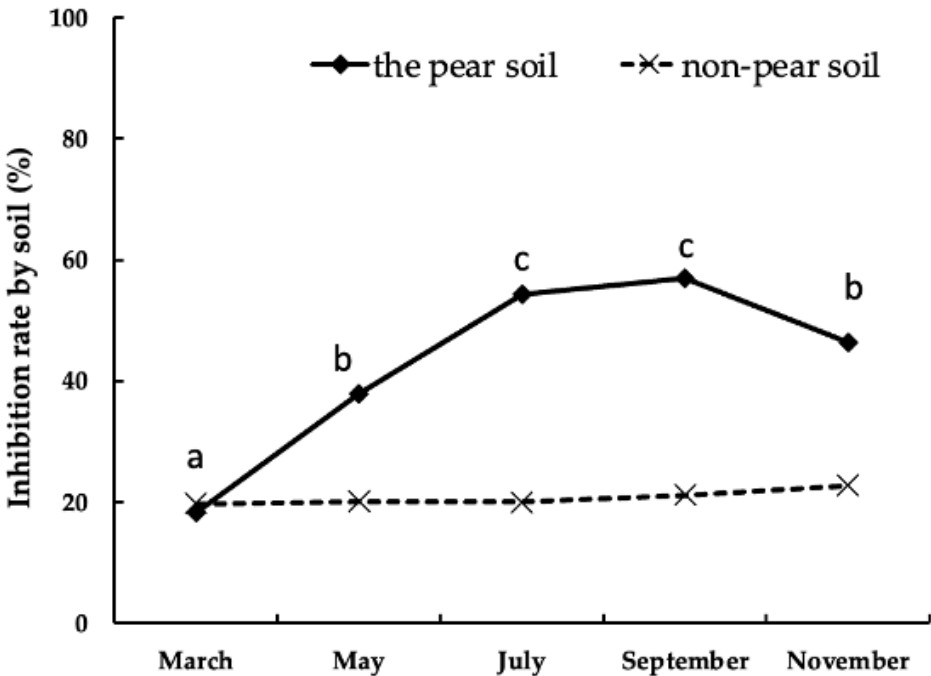

**Figure 1.** Changes over time in soil inhibition rate after planting Japanese pears. Data were arcsine transformed and analysed by Tukey–Kramer method. Lowercase letters indicate a 5% level of significance between different letters.

### 3.2. Evaluation of Growth Inhibitory Activity of Japanese Apricot and Fig Trees (Experiment 2)

In Japanese apricots, the growth inhibitory effects of the leaves, shoot barks, thick root barks, and fine roots were 90.8%, 100%, 96.8%, and 100%, respectively (Table 2). The growth inhibitory effects of the shoot pith (65.3%) and thick root pith (55.4%) were both lower than other parts of Japanese apricots. In figs, the growth inhibitory effects of the leaf (96.8%) and thick root bark (84.6%) were higher than shoot pith (57.1%), thick root pith (66.9%), and fine root (65.0%).

**Table 2.** Growth inhibitory activity of the tissue types of Japanese apricot and Fig tree.

| | Growth Inhibitory Activity of Tree (%) | |
|---|---|---|
| | Japanese Apricot | Fig |
| Leaf | 90.8 b | 96.8 b |
| Shoot bark | 100.0 b | 75.7 ab |
| Shoot pith | 65.3 a | 57.1 a |
| Thick root bark | 96.8 b | 84.6 b |
| Thick root pith | 55.4 a | 66.9 a |
| Fine root | 100.0 b | 65.0 a |
| *p*-value | <0.01 | <0.01 |

Data are arcsine transformed and analysed using the Tukey–Kramer method. Lowercase letters indicate a 5% level of significance between different letters.

### 3.3. Evaluation of the Effect of Tree Fragments Mixed with Soil on the Occurrence of Soil Sickness Syndrome in Japanese Pear (Experiment 3)

There was no difference in the growth of shoots and the main trunk diameter in the plots with soil mixed with the shoot, thick roots, and fine roots of Japanese pear compared to the unmixed plot (Table 3).

**Table 3.** The growth of Japanese pear trees in soil mixed with shoots and roots of Japanese pear trees.

| Treatments | Shoot | | | | Main Trunk Diameter (mm) |
|---|---|---|---|---|---|
| | Number (Branch/Tree) | Length (cm) | Total Elongation (m/tree) | Proximal Diameter (mm) | |
| Shoot | 6.3 | 152.8 | 9.6 | 14.5 | 30.9 |
| Thick root | 6.2 | 147.6 | 9.2 | 14.3 | 30.1 |
| Fine root | 5.4 | 178.1 | 9.6 | 16.1 | 29.6 |
| Non-treated | 7.0 | 151.2 | 10.6 | 14.3 | 30.8 |
| *p*-value | 0.44 | 0.80 | 0.40 | 0.16 | 0.30 |

Data were analysed via the Tukey–Kramer method.

In addition, regarding the raw weight of trees separated into the tissue type of the tree, there was no difference in the plot mixed with crushed shoots, thick roots, and fine roots compared to the unmixed plot (Table 4).

**Table 4.** The raw weight of trees in soil mixed with shoot and roots of Japanese pear trees.

| Treatments | Shoot | Stem | Root | Whole Tree |
|---|---|---|---|---|
| | (kg/Tree) | | | |
| Shoot | 1.29 | 0.79 | 0.72 | 2.80 |
| Thick root | 1.28 | 0.76 | 0.75 | 2.79 |
| Fine root | 1.23 | 0.75 | 0.66 | 2.64 |
| Non-treated | 1.49 | 0.80 | 0.82 | 3.11 |
| *p*-value | 0.24 | 0.75 | 0.30 | 0.29 |

Data were analysed via the Tukey–Kramer method.

### 3.4. Evaluation of the Effect of the Difference in Rootstock on the Degree of Occurrence of Soil Sickness Syndrome in Japanese Pear trees (Experiment 4)

At the time of planting, the soil inhibition rate was 62.7% in the pear soil plot, which was higher than 23.4% in the non-pear soil plot (Table 5). When the soil inhibition rate exceeds 60%, growth suppression by soil sickness syndrome will be large [9]. There was no significant difference in the main trunk diameter at the time of planting between the rootstock and the soil.

**Table 5.** Soil inhibition rate at the time of planting and main trunk diameter of Japanese pear saplings.

| Rootstock | Soil Type | Soil Inhibition Rate (%) | Main Trunk Diameter (mm) |
|---|---|---|---|
| *P. betulifolia* | Continuous cropping soil | 62.7 ** | 15.9 |
| | Non-pear soil | 23.4 | 14.9 |
| *P. pyrifolia* | Continuous cropping soil | - | 15.7 |
| | Non-pear soil | - | 14.8 |
| Interaction | Rootstock | - | ns |
| | Soil | - | ns |
| | Interaction | - | ** |

The rhizosphere soil assay method was analysed by *t*-test after arcsine transformation. The main trunk diameter was analysed by two-way ANOVA. ** Indicates a significant difference at the 1% level; ns indicates no significance; - indicates no data.

There was a significant difference between the rootstocks in the total elongation of shoots and the colour of the leaves at the end of cultivation (Table 6). In addition, there was a significant difference between the rootstock and soil in the number of leaves and the diameter of the main trunk.

**Table 6.** Difference between rootstock and soil on growth of trees.

| Rootstock | Soil Type | Shoot | | | |
|---|---|---|---|---|---|
| | | Number (Branch/Tree) | Length (cm) | Total Elongation (m/Tree) | Proximal Diameter (mm) |
| *P. betulifolia* | Continuous cropping soil | 6.0 | 61.5 | 3.8 | 8.5 |
| | Non-pear soil | 7.2 | 70.6 | 5.3 | 9.0 |
| *P. pyrifolia* | Continuous cropping soil | 6.3 | 54.7 | 3.4 | 8.1 |
| | Non-pear soil | 5.8 | 64.4 | 3.6 | 8.7 |
| Interaction | Rootstock | ns | ns | * | ns |
| | Soil type | ns | ns | ns | ns |
| | interaction | ns | ns | ns | ns |
| Rootstock | Soil Type | Leaf | | | Main Trunk Diameter (mm) |
| | | Number per Plant | | SPAD Value | |
| *P. betulifolia* | Continuous cropping soil | 156.2 | | 47.8 | 20.5 |
| | Non-pear soil | 239.6 | | 46.8 | 21.6 |
| *P. pyrifolia* | Continuous cropping soil | 135.0 | | 44.1 | 16.5 |
| | Non-pear soil | 181.2 | | 45.6 | 19.8 |
| Interaction | Rootstock | * | | * | ** |
| | Soil type | ** | | ns | ** |
| | Interaction | ns | | ns | ns |

Data were analysed by two-way ANOVA. * Indicates a significant difference at the 5% level; ** indicates a significant difference at the 1% level; ns indicates no significance.

At the end of the cultivation period, the dry weight of the leaves of the pear tree showed a significant difference between the rootstocks (Table 7). In addition, there was a significant difference between the rootstock and soil types for both the shoots and stems of the pear trees. There was also a significant difference between the rootstocks and soil regarding the whole tree and the interaction observed.

**Table 7.** Effects of rootstock and soil types on the growth (dry weight) of Japanese pear trees.

| Rootstock | Soil Type | Shoot | Leaf | Stem | Root | Whole Tree |
|---|---|---|---|---|---|---|
| | | (g/Tree) | | | | |
| *P. betulifolia* | continuous cropping soil | 88.9 | 70.5 | 176.5 | 183.6 | 519.5 |
| | non-pear soil | 141.0 | 84.7 | 214.3 | 248.1 | 688.1 |
| *P. pyrifolia* | continuous cropping soil | 76.1 | 59.7 | 132.6 | 210.9 | 479.4 |
| | non-pear soil | 88.5 | 67.3 | 141.7 | 202.8 | 500.3 |
| Interaction | rootstock | * | * | ** | ns | ** |
| | soil type | * | ns | * | ns | ** |
| | interaction | ns | ns | ns | ns | ** |

Data were analysed by two-way ANOVA. * Indicates a significant difference at the 5% level; ** indicates a significant difference at the 1% level; ns indicates no significance.

## 4. Discussion

Japanese pear trees show growth inhibitory attributes, however, the plant growth inhibitory substances have not been identified yet. To gain more insight into the growth inhibitory substance(s) in Japanese pears, the rhizosphere soil assay method was adopted to investigate the growth inhibitory activity of the pear tree over time. As a result, there were significant differences in the inhibitory effects among the various parts of Japanese pear trees in July and September. The highest inhibitory activities were observed in the thick root bark of Japanese pear trees in July (70.8%) and September (69.4%). In peaches, the major toxic cyanide was detected only in the water extract of root bark [16], and the amount of the growth inhibitory substance (amygdalin) differs depending on the sample collection time [17]. Since the bark of thick roots showed the highest inhibitory effect among the tissue types, it was inferred that a large amount of growth inhibitory substance(s) may be accumulated in the bark of thick roots. In addition, the growth inhibitory activity of the tree increased even in the tissue type where the inhibition rate was low, i.e., the thick root pith. Additionally, the growth inhibitory activity of the whole tree increased in November after defoliation. It has been reported that the sugar composition in the Japanese pear tree changes when it enters dormancy after defoliation [18]. Moreover, in apple trees, there are more growth inhibitory substances in the roots than in the leaves [19].

Changes in the growth inhibitory compounds that affect dormancy may have contributed to the observed increase in inhibitory effects in this investigation.

Furthermore, the inhibitory effects of the Japanese pear tree were compared to the growth inhibitory effects of Japanese apricots [3] and figs [20], where soil sickness syndrome is a problem. In Japanese apricots, the inhibition rates of 100%, 100%, and 96.8% for shoot bark, fine roots, and thick root bark, respectively, were higher than in other tissue types. In addition, the tissue types investigated had higher growth inhibitory activities than Japanese pears. Prunasin contained in Japanese apricot trees (especially the leaves, shoots, and roots) acts as a plant growth inhibitory substance [3]. Peach is closely related to the Japanese apricot, and it contains the growth inhibitory substance amygdalin. Amygdalin is mostly abundant in fine roots, followed by thick roots, leaves, and branches [16]. In figs, the growth inhibitory activity of the leaves was found to be 96.8%, which was higher than in other tissue types [20]. Consistent with this study, Hirai [21] reported that fig tree growth was inhibited in plots mixed with fig leaves as compared with plots treated with the shoot and other tissue types.

In this study, the growth inhibitory activity was highest in the thick roots bark of Japanese pear trees, which indicates that the growth inhibitory substances are concentrated in this part of the plant. In Japanese pear trees, arbutin accumulates in the bark and has the function of suppressing growth. Arbutin contributes to protection against phytopathogens by being converted into hydroquinone in wounds [22]. The thick root bark of Japanese pear trees was analysed, and the present authors found 2.17 mg/g·DW of arbutin in this

part of the plant (unpublished data). In the shoots of Japanese pear trees, most of the arbutin is contained in the bark [23]. The bark of the Chinese pear tree was found to contain 1.20 mg/g·FW of arbutin, which was three to five times higher than the pith [24]. Arbutin synthesized in leaves was also reported to be transmitted and accumulated in fruits, with 87.2% arbutin in the pericarp of Japanese pears [25]. In this study, the growth inhibitory activity of the trees increased during the vegetative period when the leaves were growing, which was consistent with this previous finding. The growth inhibitory substances of Japanese pear trees were previously found to be released into the soil through the roots during tree growth [10]. The results of this study show that the soil inhibition rate increased during the leaf growth stage (May to September). However, the soil inhibition rate decreased in November after the leaves stopped growing. The results suggest that the growth of the aboveground parts declined as the temperature dropped, and the production of growth inhibitory substances stopped due to the fall of leaves. The plant growth inhibitory substances in Japanese pear trees were speculated to be water-soluble [14], and thus leaching caused by rainfall or irrigation can reduce inhibitory effects. Arbutin, which acts as a biological defence for trees, is considered to be a promising candidate. Furthermore, after being synthesized in the leaves, arbutin can be released into the soil via the roots, where it can be decomposed or hydrolysed into a growth inhibitory compound, such as hydroquinone. Peach trees contain a large amount of growth inhibitory substances, such as amygdalin, which turn into harmful hydrocyanic acid when root cells are destroyed by soil microorganisms and nematodes [26]. Future research should focus on identifying and quantifying the growth inhibitory substances of the Japanese pear tree.

If the tree contains any growth inhibitory substance, residues of the tree on/in the soil can cause soil sickness syndrome. When apple tree root fragments were mixed with soil, phlorizin (a phenolic compound) was found to be generated, and the replanted apple saplings' growth was considerably inhibited [19]. However, growth inhibition caused by soil sickness syndrome does not occur in Japanese pear trees when the fine roots produced that year are mixed with non-pear soil, and pear saplings are cultivated [10]. On the other hand, since the growth inhibitory activity of Japanese pears depends on the tissue type, it is necessary to clarify the involvement of shoots and thick roots of the previous pear plant that may remain in the soil. Since the bark of the Japanese pear contains a large amount of arbutin, which is presumed to be a growth inhibitor [23], growth inhibition may occur. Although the growth inhibitory activity of the shoot is not high in the results of this study, the shoots still contained a large amount of arbutin, which may cause soil sickness syndrome as they remain in the soil. In asparagus, as the tree grows, the colour of the stored roots changes from white to tan, and the content of rutin increases from 0.24 mg/g·DW to 3.0 mg/g·DW, thus increasing growth inhibitory activity [13]. Since there are no storage roots in Japanese pear trees, the test was conducted using thick roots that had been growing for about 2 to 3 years. As a result, there was no difference in tree growth in the plots mixed with the shoot and root fragments after cultivation, and no evidence of soil sickness syndrome was observed. In a test where pear saplings were planted in a continuous cropping field of pear trees, and their growth was compared with those planted in non-pear soil, the growth of trees decreased to about 30% of those planted in the non-pear soil due to soil sickness syndrome [1]. In Japanese pear trees, attempts have been made to compost and reuse residues that were crushed with a chipper [27], and from the results of this study, soil sickness syndrome may not occur even if these are reused. In addition, in asparagus, the aerial part with less rutin, which is presumed to be a plant growth inhibitor, has been recommended as a candidate to incorporate into the soil to reduce plant residue content [28].

In peaches, research is underway to promote the cultivation of rootstock varieties using biotechnology techniques to reduce growth inhibition caused by soil sickness syndrome and pests [29]. Regarding the rootstock of Japanese pear trees, it has been clarified that the growth of *P. betulifolia* is superior to that of *P. pyrifolia* [30]; however, there is no knowledge about the strength of the soil sickness syndrome. Therefore, in this study, we conducted

a cultivation test to clarify the strength of *P. betulifolia* and *P. pyrifolia* against soil sickness syndrome. The dry weight of the shoot at the end of the cultivation test was 63.0% of the non-pear soil plot in the continuous cropping soil in *P. betulifolia*, which was more severe than the 86.0% in *P. pyrifolia*. In addition, when compared for each rootstock, differences were observed in many parameters, and the growth of *P. betulifolia* was excellent. The shoot growth of *P. pyrifolia* planted in non-pear soil was similar to *P. betulifolia* planted in continuous cropping soil, even with the effects of soil sickness syndrome. The results show that the effect of soil sickness syndrome is more pronounced in *P. pyrifolia* than in *P. betulifolia* during the initial growth stage. The soil sickness syndrome of Japanese pears is characterized by the high inhibition of pear shoot growth in the early stage of growth [10]. When the shoots are inhibited in the early growth stage, the growth of the entire tree including leaves and old branches is subsequently inhibited. The dry weights of the pear shoot, leaves, and old branches were also all inhibited in this study. On the other hand, there was no influence on the root growth of the pear tree, suggesting that growth inhibitory compounds may not directly affect the plant. There was a difference in scion growth when several varieties of Japanese pears were grafted on *P. betulifolia* rootstock in cultivation studies, but there was no difference in root growth [11]. From this, it is considered that the soil sickness syndrome may be manifested by suppressing the growth of shoots by absorbing the growth inhibitory substances in the soil from the roots of the replanted pear saplings. When Japanese pears are affected by drought stress, the yield and quality deteriorate [31]. Among the various rootstock species, *P. betulifolia* exhibits good adaptability to different environmental conditions, with high tolerance to drought, cold, and salt [12]. In addition, the occurrence of poor germination, in which flower buds die due to climate change has become a problem [32], but it has been reported that the incidence in *P. betulifolia* is higher than that of *P. pyrifolia* [33]. As a result, it is recommended to use *P. betulifolia* because the initial growth is more vigorous than *P. pyrifolia*, and it may be able to cope with climate change and tolerate the effects of soil sickness syndrome. On the other hand, growth inhibition due to soil sickness syndrome is larger than that of *P. pyrifolia*; therefore, it is necessary to take measures, such as soil dressing [34] and activated carbon treatment [6] in replanted fields.

## 5. Conclusions

In this study, we clarified using the rhizosphere soil assay, that the root bark of Japanese pear had the highest growth inhibitory activity during the growing season. The results of this study show that the root bark had the highest inhibitory activity during July, and growth inhibitory substances accumulate in the bark of the Japanese pear. Also, *P. betulifolia* as a rootstock had better initial growth in the pear-soil with soil sickness syndrome than *P. pyrifolia.* This makes *P. betulifolia* a suitable rootstock to minimize the challenge of soil sickness in Japanese pear cultivation. Future studies should focus on identifying and quantifying the growth inhibitory substances that accumulate in the thick root bark, and the effects of cultivar and rootstock combinations on the severity of soil sickness syndrome in Japanese pear cultivation.

**Author Contributions:** Conceptualization, T.T., M.O. and Y.F.; methodology, T.T., J.T. and Y.F.; software, T.T.; validation, T.T. and K.S.A.; formal analysis, T.T. and M.O.; investigation, T.T. and M.O.; resources, J.T. and Y.F.; data curation, T.T.; writing—original draft preparation, T.T.; writing—review and editing, K.S.A., Y.F. and J.T.; visualization, T.T.; supervision, Y.F.; project administration, M.O. and Y.F. All authors have read and agreed to the published version of the manuscript.

**Funding:** This work was partly supported by JST CREST Grant Number JPMJCR17O2 and JSPS KAKENHI Grant Number 26304024.

**Institutional Review Board Statement:** Not applicable.

**Informed Consent Statement:** Not applicable.

**Data Availability Statement:** Not applicable.

**Conflicts of Interest:** The authors declare no conflict of interest.

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
