# Peer review of "Plant Growth Inhibitory Activity and the Response of Different Rootstocks to Soil Sickness Syndrome in Japanese Pear Tree"

_agronomy, doi:10.3390/agronomy12092067_

Round 1

Reviewer 1 Report

1) pear tree is not Japanese pear (2 different species)

2) I suggest to use Asian pear or Nash-pear instead of Japanese pear. It is East-Asian and grown in a number of different countries

3) It is Pyrus betulifolia - and not P. betulaefolia

4) it is fig or fig-tree and not Fig (the species is Ficus carica). The same with Birchleaf and birchleaf and some more.
A focus should be set on Asparagus - which unfortunately is used as a common and a scientific name: asparagus (spears) - Asparagus officinale

5) Use, according to the Code, single quotation marks for cultivar names '...'. Please avoid variety. This is not the proper term - which actually is cultivated variety or cvar.

6) The text is frequently not precise in expression. E.g. it should be growth in acreage and not only ...growth... (where increase might be the better choice). I suggest someone else should review the paper before re-submitting.

7) Usually giving authors' names goes together with the year of publication. And you might use author-names more frequently.

8) I suggest to introduce a "Research question". To my understanding you mix it up in Introduction and Methods. (Generally - I found it hard to read and understand the paper - pls re-check and think a bit more of your readers after publication)

9) Methods and Experimental design should be worked over. And I suggest to introduce clear description of plant-material, sampling-methods and and the rhizosphere-soil-assay. 

10) L. and Sieb. et Zucc are author-names and they are not in italic (according to the Code). E.g Ficus carica L.

11) Sometimes .. roots were separated in ... might be the better choice (or root samples were divided).
I miss a clear description which way you sampled and cleaned the roots.

12) I assume you mean ...cultivation trial... (of the plantation) and not ...cultivation test...

13) the different sampling dates should be specified and explained in Methods

14) Figure 1 - Inhibition rate by soil - might be the more appropriate naming of the y-axis

15) Discussion can be improved - and more and better links to results established

16) Conclusion does not match with Results and Discussion

17) I suggest to introduce scientific names of plants already in the Abstract.

18) replant-disease and soil-sickness should be explained better in the Introduction

18) There is some self-quotation - which is ok as long as you refer to your achievements.
BUT - references can and should be improved:
E.g.  Shishu Pal Sing et al (2018); Cesarano et al (2017); Politycka Barbara (several papers); Zhao-Hui Li et al (2010); Li-Feng Huang et al (2013)

Author Response

Response to the comments of Reviewer 1

Thank you for your valuable comments. Based on your comments, we have revised as follows:

Point1:pear tree is not Japanese pear (2 different species)

Response 1: There are three types of pears: Japanese pear (Pyrus pyrifolia), Chinese pear (P. bretschneideri), and Western pear (P. communis), which are cultivated all over the world for food.

Point 2: I suggest to use Asian pear or Nash-pear instead of Japanese pear. It is East-Asian and grown in a number of different countries

Response 2: In this paper, we aimed at how to recover the continuous cropping problems of Japanese pear.

Point 3: It is Pyrus betulifolia - and not P. betulaefolia

Response 3: Thank you for your comment. We changed according to your suggestion.

 Point 4: it is fig or fig-tree and not Fig (the species is Ficus carica). The same with Birchleaf and birchleaf and some more. A focus should be set on Asparagus - which unfortunately is used as a common and a scientific name: asparagus (spears) - Asparagus officinale

Response 4: Thank you for your comments. We changed according to your suggestions.

Point 5:  Use, according to the Code, single quotation marks for cultivar names '...'. Please avoid variety. This is not the proper term - which actually is cultivated variety or cvar.

Response 5: Thank you for your comment. We changed according to your suggestion.

Point 6:  The text is frequently not precise in expression. E.g. it should be growth in acreage and not only ...growth... (where increase might be the better choice). I suggest someone else should review the paper before re-submitting.

Response 6: Soil sickness syndrome suppresses the growth of trees. Therefore, I think that it is good to use growth instead of expanding the area.

Point 7: Usually giving authors' names goes together with the year of publication. And you might use author-names more frequently.

Response 7: Thank you for your comment. We changed according to your suggestion.

Point 8: I suggest to introduce a "Research question". To my understanding you mix it up in Introduction and Methods. (Generally - I found it hard to read and understand the paper - pls re-check and think a bit more of your readers after publication)

Response 8: Thank you for your comment. I deleted the "Research question" from methodology and clarified the research purpose.

Point 9:Methods and Experimental design should be worked over. And I suggest to introduce clear description of plant-material, sampling-methods and and the rhizosphere-soil-assay. 

Response 9: The methods and experimental design have been revised to provide more clarity to the plant material, sampling methods and the rhizosphere soil assay.

Point 10: L. and Sieb. et Zucc are author-names and they are not in italic (according to the Code). E.g Ficus carica L.

Response 10: This and other issues with scientific names have been fixed as you pointed out.

Point 11: Sometimes ..roots were separated in ... might be the better choice (or root samples were divided). I miss a clear description which way you sampled and cleaned the roots.

Response 11: Thank you for your comment. We modified these as follows:.

In Experiment 1, the roots were removed from the pot and washed with water.

In Experiment 2, the roots were partly dug up and washed with water.

Point 12: I assume you mean ...cultivation trial... (of the plantation) and not ...cultivation test...

Response 12: Thank you for your comment. We deleted “cultivation trial”

Point 13: the different sampling dates should be specified and explained in Methods

Response 13: The dates for sampling have been provided for each experimental set-up.

Point 14: Figure 1 - Inhibition rate by soil - might be the more appropriate naming of the y-axis

Response 14: Thank you for your comment. Figure 1 has been revised accordingly.

Point 15: Discussion can be improved - and more and better links to results established

Response 15: Thank you for this suggestion. The discussion has been revised.

Point 16: Conclusion does not match with Results and Discussion

Response 16: The conclusion has been revised accordingly.

Point 17: I suggest to introduce scientific names of plants already in the Abstract.

Response 17: Thank you for your comment. We changed according to your suggestion.

Point 18: replant-disease and soil-sickness should be explained better in the Introduction

Response 18: Thank you for your comment. We added a citation to clarify the Introduction. We added the description of soil-sickness and cited new paper.

Point19: There is some self-quotation - which is ok as long as you refer to your achievements. But references can and should be improved:
E.g.  Shishu Pal Sing et al (2018); Cesarano et al (2017); Politycka Barbara (several papers); Zhao-Hui Li et al (2010); Li-Feng Huang et al (2013)

Response 19: Thank you for your valuable comment. We added according to your suggestion..

Reviewer 2 Report

The manuscript does not have line numbering which makes it very difficult to identify comments in the review procedure.

Introduction

1.     1. There is a lot of methodological information throughout the chapter (experiment 1, 2, 3; collecting of soil samples … ect.)

2.     There is a lack of a clearly defined research aim.

3.     A great number of scientific papers have been devoted to the problem of "soil sicknes" Despite this, the authors limited themselves to a dozen or so literature items that only partially relate to the subject matter (discussing asparagus, peaches, etc.). However, they omit the response of different rootstocks to soil disease.

Material and methods

1.     The described methodology makes it impossible to reproduce the experiment

2.     There is no description of the physical and chemical properties of the soil

3.     Why was soil taken for pots after vegetable crops?

4.     How do the authors know that the soil used in the experiment has soil sickness characteristics?

5.     Why were lettuce, Japanese apricots, and figs (see article title) used for the study?

6.     In how many repetitions were the analyses performed - in each of so many experiments.

7.     How many grams and from how many plants was the material taken for testing (leaves, bark, roots ...)

8.     Provide a detailed list of soil and other plant organ analyses performed (“The following analyses were performed: …)

9.     There are a number of repeated paragraphs in the chapter - see chapters 2.1 and 2.2

Results

1.     Table 1 Statistical analysis missing for several months

2.     What does the term "new soil" mean?

3.     The chapter contains information that has a weak connection to the topic of the study. E.g. the effect of rootstocks on growth strength or leaf color and weight ... etc.

4.     The chapter contains mutually exclusive claims. For example, the authors state: “… as a result, it is recommended to use P. pyrifolia because the initial growth is more vigorous than P. pyrifolia” (may be P.betulaefolia ??)

Discussion

The authors mostly duplicate the information in the Results section or refer to non-experimental species. 

Bibliography  

The amount of literature used is insufficient, especially for such a widely described issue as replantation disease

Author Response

Response to the comments of Reviewer 2

Thank you for your comments. According to your comments, we have revised as follows:

Point: The manuscript does not have line numbering which makes it very difficult to identify comments in the review procedure.

Response: We are sorry for this inconvenience. We added line numberings.

Introduction

Point1. There is a lot of methodological information throughout the chapter (experiment 1, 2, 3; collecting of soil samples … ect.)

Response: We removed and made clear the methodological information form the chapter.

Point 2.     There is a lack of a clearly defined research aim.

Response:We defined the research objectives at the Introduction as follows.

Point 3.     A great number of scientific papers have been devoted to the problem of "soil sicknes" Despite this, the authors limited themselves to a dozen or so literature items that only partially relate to the subject matter (discussing asparagus, peaches, etc.). However, they omit the response of different rootstocks to soil disease.

Response : Thank you for your comment. I added a paper to clarify the purpose. Soil sickness syndrome is not a disease. I added the description and the cited paper. Regarding rootstocks, there is limited study in relation to soil sickness.

Material and methods

Point1.  The described methodology makes it impossible to reproduce the experiment

Response : Thank you for your comment. The methodology has been revised.

Point 2.     There is no description of the physical and chemical properties of the soil

Response : Thank you for your comment. We added the description of the physical and chemical properties of the soil.

Point3.     Why was soil taken for pots after vegetable crops?

Response : Thank you for your comment. Soil with a history of Japanese pear cultivation shows an activity of soil sickness syndrome by suppressing the growth of the plant. Soil taken from a vegetable field with no history of Japanese pear cultivation would not exhibit soil sickness syndrome.

Point4.     How do the authors know that the soil used in the experiment has soil sickness characteristics?

Response : Thank you for your comment. when growth of pears in the continuous cropping soil decreased to 50ï¼…, we thought soil sickness problem. In addition, as described in our previous paper, plant growth inhibition measured by the rhizosphere soil method is high, we can predict the occurrence of soil sickness syndrome.

Point 5. Why were lettuce, Japanese apricots, and figs (see article title) used for the study?

Response : Lettuce was used as an indicator plant because it is susceptible to allelopathy. Soil sickness syndrome is observed in Japanese apricot and fig. These two species are used as a reference because they have a greater degree of occurrence of soil sickness syndrome than Japanese pears, and there is more knowledge.

Point6. In how many repetitions were the analyses performed - in each of so many experiments.

Response : Experiment 1 and 2, the measurement was performed 5 times in each section. Experiment 3 and 4, each of the treatments had 5 replications. Each experiment has been conducted for two years with similar results.

Point7. How many grams and from how many plants was the material taken for testing (leaves, bark, roots ...)

Response : Thank you for your comment. From three trees, 100 g each was collected for each part such as leaves and roots. We added these explanation

Point8. Provide a detailed list of soil and other plant organ analyses performed (“The following analyses were performed: …)

Response : Thank you for your comment. We describing these details in the method.

Point9. There are a number of repeated paragraphs in the chapter - see chapters 2.1 and 2.2

Response:Thank you for your comment. We have removed the overlap as much as possible.

Results

Point1.     Table 1 Statistical analysis missing for several months

Response1:The survey was conducted at the timing when the pear growth stage changes (every two months).

Point2.     What does the term "new soil" mean?

Response:Thank you for your comment. This means ‘the soil that Japanese pear never grown’. We changed the expression as this.

Point3.     The chapter contains information that has a weak connection to the topic of the study. E.g. the effect of rootstocks on growth strength or leaf color and weight ... etc.

Response:Thank you for your comment. I (Toya) am a specialist of Japanese pear production working at Fruit Research Station at local Government for 20 years, and in my experience, soil sickness syndrome that the growth of pear trees declining, is strongly related to the growth of new shoots and leaves.

Point4.     The chapter contains mutually exclusive claims. For example, the authors state: “… as a result, it is recommended to use P. pyrifolia because the initial growth is more vigorous than P. pyrifolia” (may be P.betulaefolia ??)

Response:Thank you for your valuable comment.. These are our error. We have changed according to your suggestion.

Discussion

Point: The authors mostly duplicate the information in the Results section or refer to non-experimental species. 

Response : Thank you for your comment. We have deleted some information from the results section.

Bibliography  

Point: The amount of literature used is insufficient, especially for such a widely described issue as replantation disease

Response : Thank you for your comment. We have increased the number of citations as much as possible. However, there are not so many literatures on the replant problems of Japanese pear cultivation.

Reviewer 3 Report

The authors have presented a valuable topic especially to the practice. Although the manuscript is structured very well it can benefit from several small improvements.

In the Introduction section authors can include several sentences commenting on the soil sickness syndrome in general and the contemporary approaches used in the agricultural sector.

When the aim of the study is defined it seems reasonable to define the application of the outcomes of the current study.

The statistical analysis of the data  can be presented separately in MM section because it will give a clearer vision for the reader. At the moment, some reader may assume that statistical evaluation is lacking in this study.

Just in case, the manuscript should be double checked for minor mistakes.

Author Response

Response to Reviewer 3 Comments

Thank you for the time you have taken to review this manuscript. Based on your feedback, we have revised it. thank you.

Point1: In the Introduction section authors can include several sentences commenting on the soil sickness syndrome in general and the contemporary approaches used in the agricultural sector.

Response1: Thank you for your comments. We added the sentences commenting on the soil sickness syndrome and the research of Japanese pear in the Introduction section.

Point2: When the aim of the study is defined it seems reasonable to define the application of the outcomes of the current study.

Response 2: Thank you for your encouraging comment. We added the outcome of this research at the conclusion and summary as: “We think the results in this study and application of rhizosphere soil assay will contribute to reduce the continuous cropping problems of Japanese pear.”

Point3: The statistical analysis of the data  can be presented separately in MM section because it will give a clearer vision for the reader. At the moment, some reader may assume that statistical evaluation is lacking in this study.

Response 3: Thank you for your comment. We modified the explanation for Statistical analysis at 2-5.

Round 2

Reviewer 2 Report

Thank you to the authors for considering the comments.